# Preferential assembly of heteromeric kainate and AMPA receptor amino terminal domains

**Huaying Zhao[1]\*, Suvendu Lomash[2†], Sagar Chittori[2‡], Carla Glasser[2§], Mark L Mayer[2†]\*, Peter Schuck[1]\***

[1]Dynamics of Molecular Assembly Section, Laboratory of Cellular Imaging and Macromolecular Biophysics, National Institute of Biomedical Imaging and Bioengineering Institutes of Health, National Institutes of Health, Bethesda, United States; [2]Laboratory of Cellular and Molecular Neurophysiology, Porter Neuroscience Research Center, National Institute of Child Health and Human Development, National Institutes of Health, Bethesda, United States

**\*For correspondence:**
zhaoh3@mail.nih.gov (HZ);
mark.mayer@nih.gov (MLM);
schuckp@mail.nih.gov (PS)

**Present address:** †Molecular Physiology and Biophysics Section, National Institute of Neurological Disorders and Stroke, National Institutes of Health, Bethesda, United States; ‡Laboratory of Cell Biology, Center for Cancer Research, National Cancer Institute, National Institutes of Health, Bethesda, United States; §Laboratory of Molecular and Cellular Neurobiology, National Institute of Mental Health, National Institutes of Health, Bethesda, United States

**Competing interests:** The authors declare that no competing interests exist.

**Abstract** Ion conductivity and the gating characteristics of tetrameric glutamate receptor ion channels are determined by their subunit composition. Competitive homo- and hetero-dimerization of their amino-terminal domains (ATDs) is a key step controlling assembly. Here we measured systematically the thermodynamic stabilities of homodimers and heterodimers of kainate and AMPA receptors using fluorescence-detected sedimentation velocity analytical ultracentrifugation. Measured affinities span many orders of magnitude, and complexes show large differences in kinetic stabilities. The association of kainate receptor ATD dimers is generally weaker than the association of AMPA receptor ATD dimers, but both show a general pattern of increased heterodimer stability as compared to the homodimers of their constituents, matching well physiologically observed receptor combinations. The free energy maps of AMPA and kainate receptor ATD dimers provide a framework for the interpretation of observed receptor subtype combinations and possible assembly pathways.
DOI: https://doi.org/10.7554/eLife.32056.001

## Introduction

At excitatory synapses in the brain of vertebrates, signal transduction is mediated by a large family of glutamate receptor ion channels (iGluRs) encoded by 18 iGluR subunit genes. These membrane proteins assemble as tetramers, with coassembly between subunits limited to four major families named AMPA, kainate, NMDA and delta receptors. Within each family, there are four AMPA receptor subunits, five kainate receptor subunits, seven NMDA receptor subunits, and two delta receptor subunits, which assemble in different combinations to generate various homomeric or heteromeric species in vivo. Functional experiments have shown that individual iGluR subtypes and their assemblies exhibit unique gating characteristics (*Traynelis et al., 2010*). Key insights into the formation of tetrameric receptors were provided by iGluR crystal and cryo-EM structures. These revealed tetramers with a modular architecture formed by a layered assembly of extracellular amino terminal (ATD), ligand binding (LBD), transmembrane ion channel (TMD), and cytoplasmic (CTD) domains (*Sobolevsky et al., 2009*; *Schauder et al., 2013*; *Karakas and Furukawa, 2014*; *Lee et al., 2014*; *Meyerson et al., 2014*). The ATD and LBD, both of which combine as dimers with two-fold symmetry, can be genetically excised and individually expressed as soluble proteins; when crystallized, the soluble proteins form dimeric structures identical to those found in intact receptors (*Mayer, 2011*; *Furukawa, 2012*), motivating the study of their assembly using biophysical techniques.

To gain insight into the energetics and specificity of iGluR assembly, multiple studies have used analytical ultracentrifugation (AUC) to examine the oligomerization of both the ATD and LBD expressed as soluble proteins, allowing analysis of both self-association and hetero-association processes in solution over a wide range of affinities. These studies revealed that the equilibrium dissociation constant ($K_D$) for dimer assembly by the LBD is too weak to reliably measure (*Furukawa et al., 2005*; *Weston et al., 2006*), except for proteins in which mutations in the LBD dimer interface strengthen the interaction between subunits (*Sun et al., 2002*; *Furukawa et al., 2005*; *Chaudhry et al., 2009*), indicating a $K_D$ for wild type iGluR LBDs in the mM range. By contrast, the ATDs of iGluRs form dimers at low nM to low µM concentrations (*Jin et al., 2009*; *Kumar et al., 2009*; *Karakas et al., 2011*; *Rossmann et al., 2011*), suggesting that ATD oligomerization plays a major role in controlling iGluR assembly. In support, dimeric GluA2 assembly intermediates have been resolved by single-particle EM studies after synchronized protein expression, showing closely apposed ATD domains (*Shanks et al., 2010*). Further, electrophysiological studies with ATD mutants have shown that ATD interactions are critical for the assembly of functional heteromeric receptors (*Kumar et al., 2011*; *Rossmann et al., 2011*). On the other hand, additional important contributions in iGluR assembly are likely to arise from other interactions, including the transmembrane domain (*Gan et al., 2016*), and auxiliary proteins (*Jackson and Nicoll, 2011*; *Sheng et al., 2017*). Currently the energetic contributions and importance of these domains relative to that of the ATD are not clear.

The study of iGluR high-affinity homo- and hetero-associations was greatly facilitated in the last decade by the introduction of a commercial fluorescence detection system (FDS) for analytical ultracentrifugation (*MacGregor et al., 2004*; *Schuck et al., 2015*). In pioneering work by *Rossmann et al. (2011)* this was applied to study the assembly pathways of AMPA receptor ATDs, unexpectedly indicating GluA2 homodimers to be of only slightly lower stability than the GluA2/GluA1 and GluA2/GluA3 heterodimers, the major species found in mammalian neurons (*Wenthold et al., 1996*). However, the results were impacted by technical issues resulting from non-specific interactions due to the hydrophobic FAM label attached as an extrinsic fluorophore (*Rossmann et al., 2011*; *Zhao et al., 2012*), compounded by the lack of precise analytical tools for the quantitative study of data from FDS sedimentation velocity (FDS-SV) experiments. Using model systems, in subsequent studies we have systematically addressed these issues, identifying DyLight 488 and EGFP as labels better suited for FDS experiments on iGluR ATDs, and have developed analytical procedures and controls that allow measurement of protein interactions with low pM equilibrium dissociation constants (*Zhao et al., 2013d*, *2013c*, *Zhao et al., 2014*; *Chaturvedi et al., 2017a*). Applied to GluA2 homodimerization (*Zhao et al., 2012*, *2013d*), this revealed approximately an order of magnitude weaker interaction than initially reported (*Rossmann et al., 2011*), reinforcing the importance of obtaining accurate data to quantitatively study the preferential assembly of iGluR ATDs.

Surprisingly, relatively little is known about the interactions of kainate receptor (KaiR) ATDs. Kainate receptors are encoded by two gene families, GluK1-GluK3, for which the ATDs share 68–75% amino acid sequence identity; and GluK4-GluK5 which share 63% identity, but exhibit only 25–27% identity with GluK1-GluK3. GluK1-GluK3 form functional homomeric receptors but GluK4 and GluK5 are obligate heteromers which form functional channels only when coexpressed with GluK1-GluK3 (*Perrais et al., 2010*). Experiments with recombinant receptors and genetically targeted mice also reveal that GluK1-GluK3 can coassemble to form functional heteromeric KaiRs (*Cui and Mayer, 1999*; *Mulle et al., 2000*; *Veran et al., 2012*) but at present it is not known whether like AMPA receptors these subunits assemble in preferred combinations. Prior work on the assembly of kainate receptor ATDs has been limited to measurements of self-association for GluK2 (previously referred to as GluR6), with a $K_D$ of 250 nM, and GluK5 (previously referred to as KA-2) with a $K_D$ of 350 µM, and their heterodimer assembly, with a $K_D$ of 11 nM (*Kumar et al., 2011*). These values were determined using sedimentation velocity analysis with conventional absorbance and interference optics, and for the GluK2/GluK5 heterodimer assembly we were unable to work at sufficiently low protein concentrations to obtain full dissociation of the complex (*Kumar et al., 2011*). Currently, we do not know if GluK4, like GluK5, forms preferential high affinity heterodimer assemblies with GluK2, and whether the GluK4 and GluK5 ATDs form high affinity heteromeric assemblies with ATDs from other members of the GluK1-GluK3 KaiR family.

Therefore, here we embark on a systematic study of the affinities of all KaiR homo- and hetero-dimers, and in addition use improved methods to revisit the interactions of AMPA receptor ATDs in a systematic survey of their homomeric and heteromeric binding affinities. We find distinct patterns of ATD assembly that differ between AMPA and kainate receptor families, with dissociation equilibrium constants spanning several orders of magnitude, revealing clearly energetically favored homomeric and heteromeric assemblies. In addition, a more detailed analysis of sedimentation profiles for some species suggests that there are intriguing differences in their binding kinetics underlying the differences in affinity, which may also contribute to modulate assembly pathways.

## Results

### SV analysis of kainate receptor ATD assemblies

Because mixtures of KaiR ATDs exhibit simultaneous and competitive homo-dimerization and hetero-dimerization equilibria, the strategy used for their analysis was to characterize their self-association first, such that the homo-dimerization $K_D$ can be fixed in the analysis of mixtures, which then can reveal the affinity of hetero-dimerization.

#### Self-association of KaiR ATDs

Based on the results of preliminary experiments, GluK1, GluK2, and GluK3 ATDs exhibit high-affinity self-association requiring the use of fluorescence detection in SV (FDS-SV). These KaiR ATDs were purified and labeled with DyLight 488 as described in Methods and Materials. Avoiding the pitfalls of FAM, DyLight 488 has been shown previously to be an inert tag (*Zhao et al., 2013d*). For GluK1 and GluK2 we performed tracer titration experiments in which a series of concentrations of unlabeled protein was added to a low concentration of the labeled sample, shifting the sedimentation profile from monomer to dimer species. For GluK3, which expressed less well than other KaiR species, we performed a dilution series of the fluorescently labeled sample. In contrast to GluK1-3, only weak self-association is shown by GluK4 and GluK5; therefore unlabeled GluK4 and GluK5 could be studied by SV with conventional absorbance optics.

Sedimentation coefficient distributions $c(s)$ for these experiments are shown in *Figure 1* and reveal three distinct patterns. GluK1 and GluK2 share the common pattern of a major peak, which shifts to higher sedimentation coefficients as the concentration increases over the range of 0.2 nM to 40 μM for GluK1 (Panel A) and 1 nM to 500 nM for GluK2 (Panel B), respectively. By contrast, although no concentration dependent shift in peak position is observed for either GluK3 (Panel C) or GluK4 (Panel D) the sedimentation coefficients for these species are strikingly different. For GluK3 the peak at ~5.8 S is consistent with an essentially pure dimer population over a concentration range from 50 pM to 1.2 μM, while for GluK4 the peak at ~3.8 S is consistent with an essentially pure monomer population over a concentration range from 0.3 to 29 μM. This indicates that under the current experimental conditions, the GluK3 ATD has an extremely high affinity for homo-dimerization ($K_D$ < 50 pM), while GluK4 has very low affinity ($K_D$ > 1 mM), even weaker than that previously determined for GluK5 ($K_D$ = 350 μM [*Kumar et al., 2011*]). To determine the $K_D$ for homo-dimerization of GluK1 and GluK2 we performed integration from 2 to 8 S of the $c(s)$ traces for each concentration shown in *Figure 1A/B* to calculate the signal-weighted average sedimentation coefficients ($s_w$); the resulting $s_w$ isotherms were fit using a model for a monomer-dimer equilibrium; this gave a $K_D$ of 1.2 μM (95% CI: 0.8–1.7 μM) for GluK1 (Panel E) and 160 nM (95% CI: 70–350 nM) for GluK2 (Panel F).

#### Hetero-association of KaiR ATDs

Hetero-association between KaiR ATDs was studied using FDS-SV titration series experiments in which an unlabeled ATD component was added at different final concentrations to a low concentration of a second ATD binding partner carrying the fluorescent label. Because at the lowest concentrations studied, GluK3 did not dissociate into monomers, we did not study its association with other KaiR ATDs. For the combination GluK1/GluK2 we used 1 nM labeled GluK1, mixed with GluK2 ranging from 0.3 nM to 20 μM (*Figure 2*). The $c(s)$ distributions shown in Panel A reflect a competitive interaction between formation of GluK1 homodimers with a $K_D$ of 1.2 μM, GluK2 homodimers with a $K_D$ of 163 nM, and GluK1/GluK2 heterodimers. At a concentration of 1 nM, GluK1 forms an

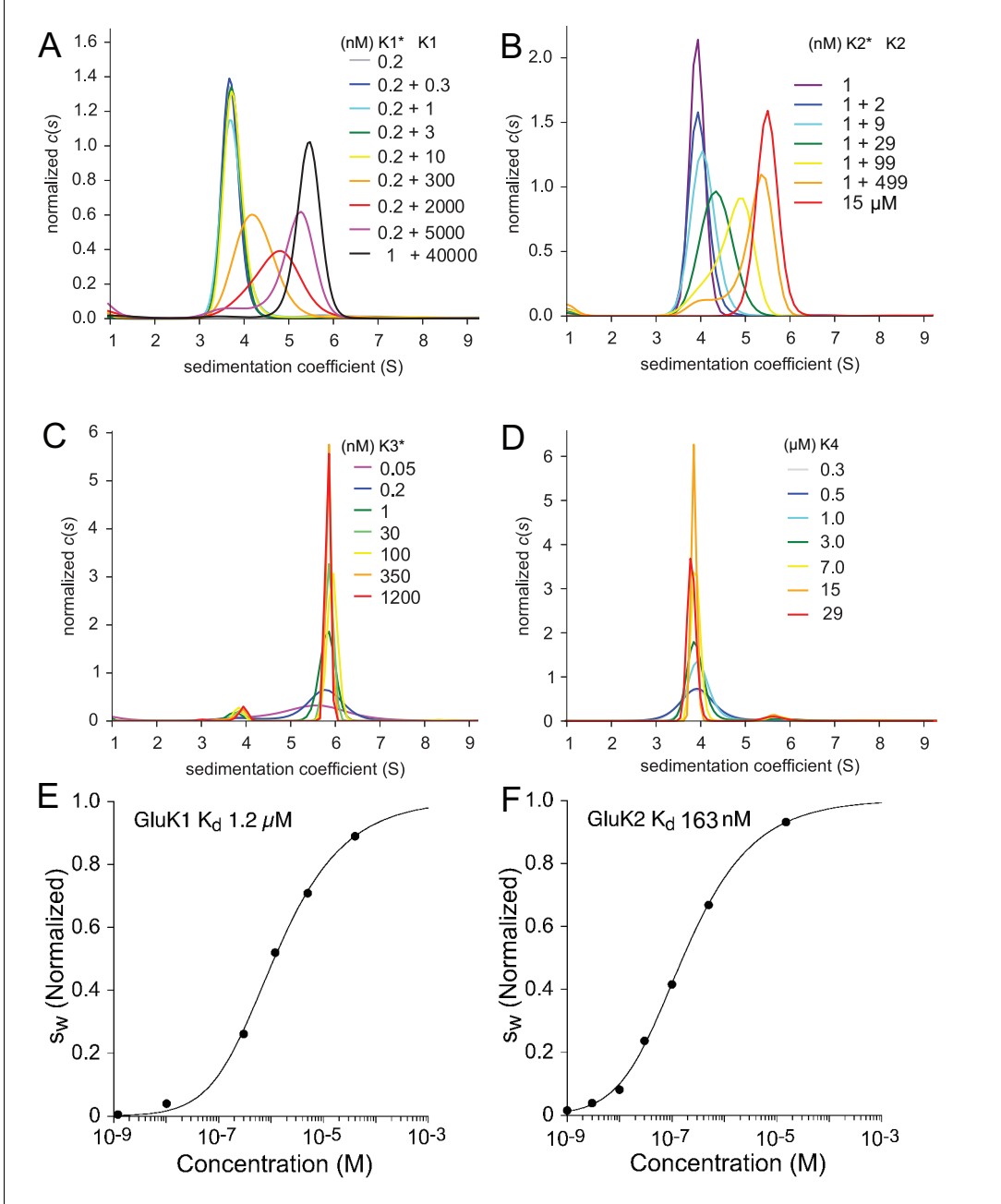

**Figure 1.** Sedimentation velocity analysis of the homo-dimerization of GluK1 (**A**), GluK2 (**B**), GluK3 (**C**), and GluK4 (**D**). Panel A: Sedimentation coefficient distributions $c(s)$ derived from FDS-SV data of samples with 0.2 nM or 1 nM DyLight 488-labeled GluK1 mixed with 0.3 nM - 40 μM unlabeled GluK1 at the concentrations indicated in the legend (with the asterisk indicating the molecule carrying the fluorophore). Panel B: $c(s)$ traces for an analogous titration series of 1 nM DyLight 488-labeled GluK2 with 2–499 nM unlabeled GluK2 using fluorescence detection, and a 15 μM sample of unlabeled GluK2 determined by absorbance detection at 280 nm. Panel C: $c(s)$ distributions from fluorescence data from DyLight 488-GluK3 at 50 pM - 1.2 μM. Panel D: Analogous $c(s)$ distributions of 0.3–29 μM unlabeled GluK4 detected by absorbance at 280 nm. Panels E and F: Isotherms of signal-weighted average sedimentation coefficients $s_w$ of GluK1 and GluK2, derived from the $c(s)$ distributions in A and B, respectively, and best-fit isotherms of a monomer-dimer model with refined $K_D$ of 1.2 μM for GluK1 *E* and 160 nM for GluK2 *F*, respectively. For better visual comparison, $s_w$ is normalized to assign a value of 1.0 to the range from monomer to dimer $s$-value.

DOI: https://doi.org/10.7554/eLife.32056.003

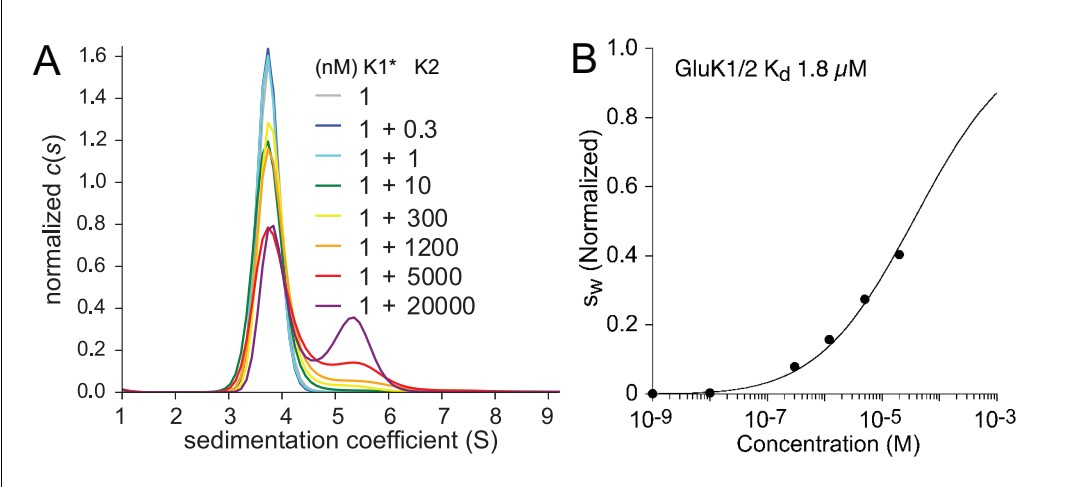

**Figure 2.** Hetero-dimerization of GluK1 and GluK2 ATD. Panel A: Sedimentation coefficient distribution $c(s)$ based on FDS-SV data of 1 nM DyLight 488-labeled GluK1 mixed with 0.3–20 µM unlabeled GluK2. A GluK1 control sample without GluK2 (grey) is virtually superimposed by those for 0.3 nM (black) and 1 nM (cyan) GluK2. Panel B: $s_w$ isotherm (normalized to assign a value of 1.0 to the range from monomer to dimer $s$-value) after integration of $c(s)$ traces (circles) and fit with a model for competitive homo- and hetero-association, fixing the two previously measured homo-dimerization constants of GluK1 and GluK2 while refining the $K_D$ for GluK1/GluK2 heteromers to the best-fit value of 1.8 µM.
DOI: https://doi.org/10.7554/eLife.32056.004

essentially pure monomer population, and thus in the titration series with unlabeled GluK2 the appearance of a peak at ~5.5 S indicates formation of the GluK1/GluK2 heterodimer. This is clearly visible at GluK2 concentrations of 5 and 20 µM (red and purple lines, respectively) but at the highest GluK2 concentration studied there is a still significant amount of GluK1 monomer present. This reflects both the high affinity of the GluK2 ATD homodimer reaction, which strongly competes with the formation of GluK1/GluK2 heterodimers, and practical limitations on the upper limit of GluK2 concentrations that can be prepared from the available samples. The $s_w$ isotherm calculated from this experiment (Panel B) was fit with a coupled equilibrium model, in which the $K_D$ and $s$-values for the two homo-dimerization reactions were fixed at values determined from independent measurements (*Figure 1* and *Table 1*), while the $K_D$ and $s$-value for the heterodimer are refined. The best-fit heterodimer $K_D$ of 1.8 µM (95% CI: 1.5–2.2 µM) indicates that GluK1 and GluK2 do not preferentially coassemble to form heterodimers; in fact, this interaction is weaker than the homo-dimerization reactions of either of its components, implying a preference for homodimer formation by KaiRs from the GluK1-3 gene family members.

To study heterodimer association for the other KaiR combinations, unlabeled constructs were titrated into labeled binding partners. The $c(s)$ distributions for the these mixtures, shown in *Figure 3*, exhibit similar features for all explored pairs, with a major peak shifting to higher $s$-values with

**Table 1.** $K_D$-values for kainate receptor amino-terminal domain dimer assembly.

|  | DL-GluK1 | DL-GluK2 | DL-GluK3 | DL-GluK4 | DL-GluK5 |
|---|---|---|---|---|---|
| UL GluK1 | 1.2 µM [0.8, 1.7] |  |  |  |  |
| UL GluK2 | 1.8 µM [1.5, 2.2] | 163 nM [113, 235] |  |  | 9.4 nM [8, 11] |
| DL GluK3 |  |  | <50 pM* |  |  |
| UL GluK4 | 146 nM [116, 184] | 57 nM [46, 71] |  | >1 mM$ |  |
| UL GluK5 | 18 nM [15,22] | 12 nM [10, 14] |  |  | 350 µM% |

Values in brackets indicate 95% confidence limits. Abbreviations: UL, Unlabeled protein; DL, DyLight 488-labeled protein; EGFP, EGFP fusion protein. *$K_D$ determination from FDS-SV dilution series. $$K_D$ determined from SV ABS data. %$K_D$ previously determined from SV ABS data for unlabeled GluK5 (**Kumar et al., 2011**).
DOI: https://doi.org/10.7554/eLife.32056.002

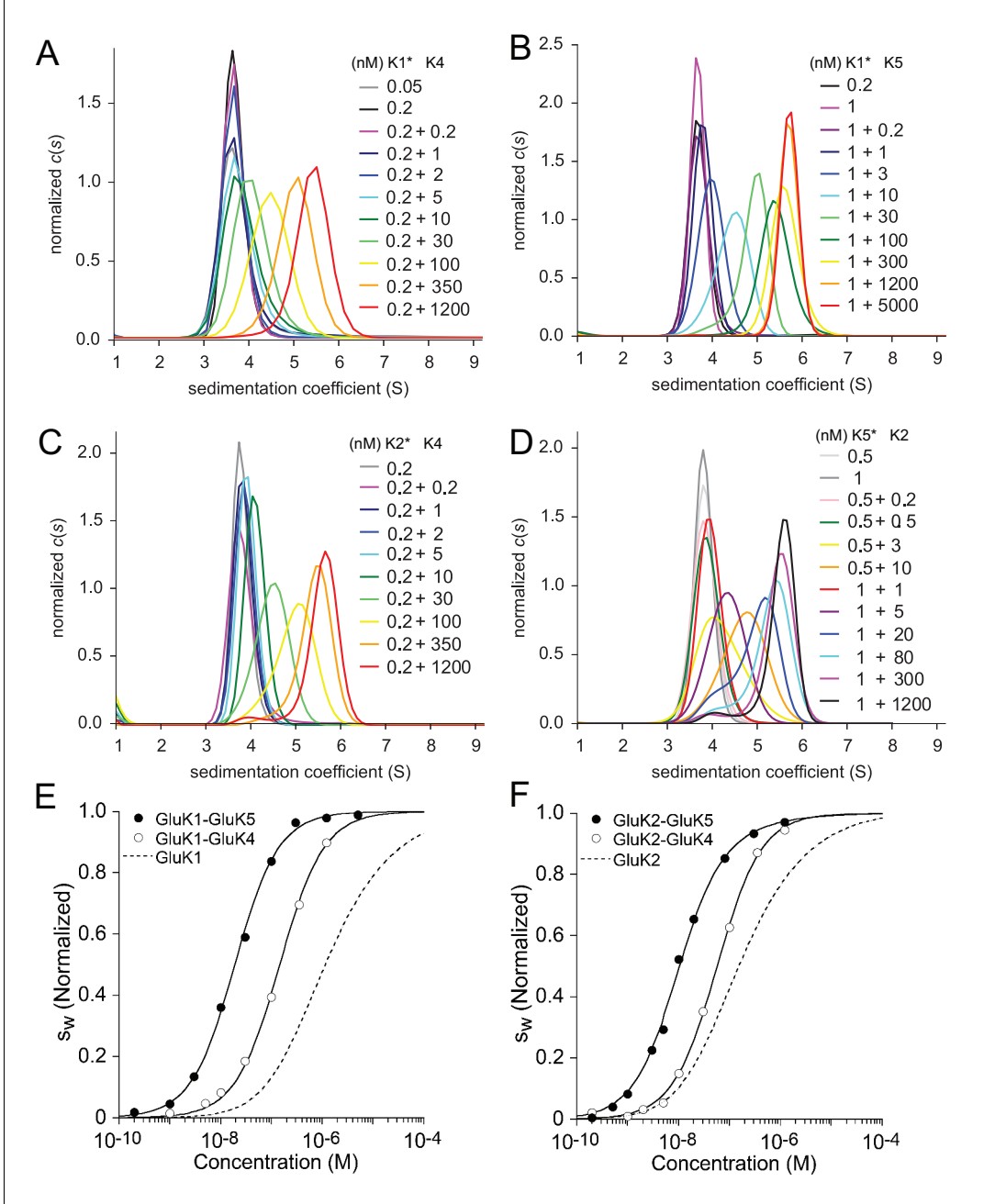

**Figure 3.** Sedimentation velocity analysis of the hetero-dimerization of GluK1 and GluK2 with GluK4 and GluK5.  Panel A: Sedimentation coefficient distributions $c(s)$ determined from analysis of FDS-SV data for titration of 50–200 pM DyLight 488-labeled GluK1 with 0.2 nM - 1.2 µM unlabeled GluK4. Panel B: $c(s)$ distributions for an analogous titration series of 200 pM or 1 nM DyLight 488-labeled GluK1 with 0.2 nM - 5 µM unlabeled GluK5. Panel C: $c(s)$ distributions from analysis of FDS-SV data for titration of 200 pM DyLight 488-labeled GluK2 with 0.2 nM - 1.2 µM unlabeled GluK4. Panel D: Analogous $c(s)$ distributions from FDS-SV data for the titration of 0.5 nM or 1 nM DyLight 488-labeled GluK5 with 0.2 nM - 1.2 µM unlabeled GluK2. Panel E: Isotherms of $s_w$-values for the titration series of GluK1 with either GluK4 or GluK5, fit with a coupled equilibrium model for GluK1/GluK4 or GluK1/GluK5 heterodimer assembly competitive with homo-dimerization of both binding partners, leads to a best-fit heterodimer $K_D$ of 146 nM (95% CI: 116–184 nM) for GluK1/GluK4 and 18 nM (95% CI: 15–22 nM) for GluK1/GluK5. For comparison, the dashed line shows the isotherm for homomeric GluK1. $s_w$ is normalized to assign 1.0 for the range from monomer to dimer $s$-value. Panel F: Analogous to $E$, isotherms of $s_w$-values for the titration series of GluK2 with either GluK4 or GluK5, fit with a coupled equilibrium model for GluK2/GluK4 or GluK2/GluK5 heterodimer assembly competitive with homo-dimerization of both binding partners, lead to a heterodimer $K_D$ of 57 nM (95% CI: 46–71 nM) for GluK1/GluK4 and 12 nM (95% CI: 6–17 nM) for GluK1/GluK5; the dashed line shows the isotherm for homomeric GluK2.
DOI: https://doi.org/10.7554/eLife.32056.005

increase in concentration. For all combinations studied, at the highest concentration of the unlabeled protein (1.2 μM or 5 μM), the main peak sediments at ~5.6 S close to the expected $s$-value for the dimer, indicating nearly full binding saturation; conversely, at the lowest concentrations studied (0.2–0.5 nM) the labeled species sediments at ~3.8 S close to the expected $s$-value for the monomer *Figure 3*. It can be discerned that transition between monomer and dimer species in different hetero-pairs occurs at higher concentrations for GluK1/GluK4 (*Panel A*) and GluK2/GluK4 (*Panel C*) than for GluK1/GluK5 (*Panel B*) and GluK2/GluK5 (*Panel D*). This difference can be clearly identified by comparing the peak position of the sample at X nM +10 nM (where X is 0.2, 0.5 or 1 nM of the particular labeled species, with 10 nM unlabeled binding partner added): For example, for GluK1/GluK4, 0.2 nM +10 nM (dark green curve in *Panel A*) the peak is located at ~3.8 S, indicating a mainly monomer population, while for GluK2/GluK5 0.5 nM + 10 nM (orange curve in *Panel D*) the peak is located at ~4.7 S, indicating a shift towards the dimer species.

To determine the $K_D$ for KaiR heterodimer ATD assembly by GluK1 and GluK2 with GluK4 and GluK5, the $c(s)$ traces for each concentration were integrated between 2 and 8 S to generate $s_w$ binding isotherms (*Figure 3E/F*). These were fit with a coupled equilibrium model in which the $K_D$ and $s$-values for the two homo-dimerization reactions were fixed at values determined from independent measurements as described above (*Figure 1* and *Table 1*), with the $K_D$ and $s$-value for the heterodimer fit as free parameters. In summary, within the kainate receptor gene families, GluK1/GluK2 ATD heterodimers exhibit a $K_D$ of 1.8 μM, which is similar in affinity to the assembly of GluK1 homodimers ($K_D$=1.2 μM), but the GluK1/GluK2 heterodimer is 10-fold less stable than the GluK2 homodimer ($K_D$=163 nM). By contrast, the GluK4 and GluK5 ATDs both show a strong preference for heterodimer assembly with GluK1 and with GluK2, but with subunit dependent differences in affinity *Table 1*. For GluK1 (*Figure 3E*), we observed an 8-fold lower affinity for coassembly with GluK4 (GluK1/GluK4 $K_D$ = 146 nM; 95% CI: 116–184 nM), compared to coassembly with GluK5 (GluK1/GluK5 $K_D$ = 18 nM; 95% CI: 15–22 nM). For GluK2 (*Figure 3F*), we observe a 5-fold lower affinity for coassembly with GluK4 (GluK2/GluK4 $K_D$ = 57 nM; 95% CI: 46–71 nM), compared to coassembly with GluK5 (GluK2/GluK5 $K_D$ = 12 nM; 95% CI: 6–17 nM).

## SV analysis of AMPA receptor ATD interactions

All AMPA receptor ATDs undergo high-affinity self- and/or hetero-dimerization, and thus require fluorescently labeled versions for all members. GluA2 ATD was chemically labeled with DyLight 488 as previously described (*Zhao et al., 2013d*), whereas GluA1 and GluA4 ATDs were expressed as EGFP fusion proteins, which is equally inert with regard to interactions as DyLight 488 (*Zhao et al., 2013d*). Because the EGFP-GluA1 and EGFP-GluA4 ATDs expressed well in HEK cells, the comparatively large quantity of materials available allowed us to employ a dilution series strategy, taking advantage of the high signal/noise ratio of the data from samples at higher concentrations.

### Self-association of AMPA ATDs

As described above for kainate receptors, we first determined the $K_D$ and sedimentation coefficients for self-association of AMPA receptors ATDs, and then used these as constraints for analysis of heteromeric assemblies. Because we had previously studied homodimer formation by the GluA2 and GluA3 ATDs (*Zhao et al., 2012*, *2013d*), in the present study these measurements were only performed for GluA1 and GluA4. The $c(s)$ distributions for the dilution series of GluA1 and GluA4 ATD-EGFP fusion proteins are shown in *Figure 4A/B*. For GluA1, a minor peak at 2.5 S and two major peaks at 4.4 S and ~6–6.5 S can be discerned. From hydrodynamic scale relationships the $s$-values for the GluA1 ATD-EGFP monomer and dimer are expected to be ~4–5 S and ~6–7 S, respectively. The 2.5 S peak matches the value determined previously for EGFP (*Zhao et al., 2013c*), and thus it is likely that free EGFP was present in this preparation, perhaps due to proteolytic cleavage of the fusion protein; consistent with this the amplitude of the 2.5 S peak varied between preparations. In any event, due to hydrodynamic separation from the GluA1 monomer and dimer peaks, we were able to unambiguously exclude the free EGFP signal from the integration of $c(s)$ distributions and the subsequent isotherm analysis. From the overlay of the $c(s)$ traces it can be visually discerned that the monomer to dimer transition occurs at lower concentrations for GluA1 than GluA4. This is reflected in the quantitative analysis of the $s_w$ isotherms after integration of $c(s)$ from 3 to 8 S, shown in *Figure 4C/D* together with previously determined data for GluA2 (*Zhao et al., 2013d*) and GluA3

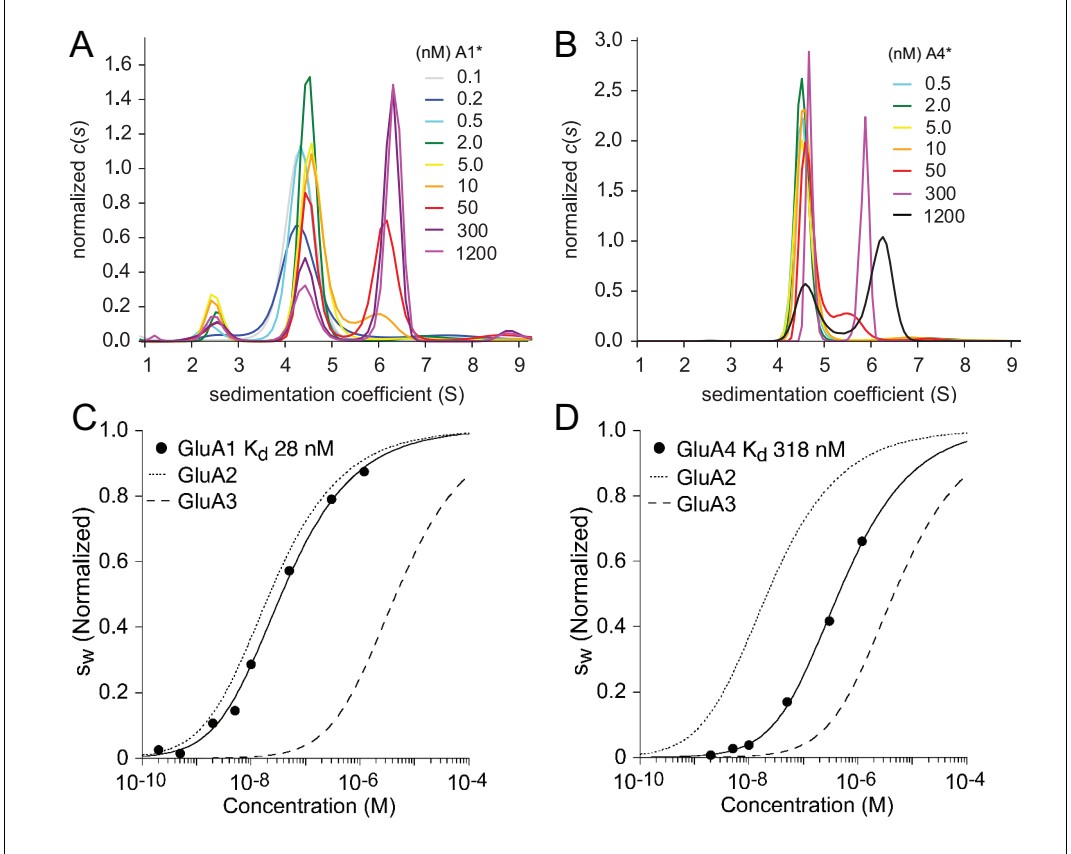

**Figure 4.** Homomeric assembly of AMPA receptor ATDs. Panels A and B: Sedimentation coefficient distributions $c(s)$ derived from FDS-SV data of a dilution series of 100 pM - 1.2 μM EGFP-GluA1 (**A**) and 500 pM - 1.2 μM EGFP-GluA4. Panels C and D: Isotherms of $s_w$ for GluA1 (**C**) or GluA4 (**D**) fit with a model for monomer-dimer association with a best-fit $K_D$ for GluA1 of 28 nM (95% CI: 14–58 nM) and for GluA4 of 318 nM (95% CI: 145–730 nM). For comparison, the dotted and dashed lines show isotherms for GluA2 and GluA3, respectively. $s_w$ is normalized to assign a value of 1.0 to the range from monomer to dimer $s$-value.

DOI: https://doi.org/10.7554/eLife.32056.006

(**Zhao et al., 2012**). The $K_D$-values were 28 nM (95% CI, 14–58 nM) for GluA1 and 318 nM (95% CI, 150–730 nM) for GluA4. As shown in the diagonal of **Table 2**, for AMPA receptor ATDs self-association covers a wide range of binding strength, with the $K_D$ for GluA1 and GluA2 the strongest (28 and 20 nM, respectively), intermediate for GluA4 (318 nM), and weakest for GluA3 (5.2 μM).

**Table 2.** $K_D$-values for AMPA receptor amino-terminal domain dimer assembly.

|  | EGFP-GluA1 | DL-GluA2 | UL-GluA3 | EGFP-GluA4 |
|---|---|---|---|---|
| **UL GluA1** | 28.4 nM [14, 58] |  |  |  |
| **UL GluA2** | 2.9 nM [1.6, 5.6] | 21.1 nM[*] [17, 27] |  | 32.6 nM [19, 59] |
| **UL GluA3** | 8.3 nM [6, 12] | 1.3 nM [1.0, 1.7] | 5.2 μM[#] [1.7, 14] | 91 nM [60,137] |
| **UL GluA4** | 62 nM [15, 218] |  |  | 318 nM [145, 730] |

Values in brackets indicate 95% confidence limits. Abbreviations: UL, Unlabeled protein; DL, DyLight 488-labeled protein; EGFP, EGFP fusion protein. [*]Similar estimates were obtained separately in previous studies for GluA2 ATD: 24 nM in (**Zhao et al., 2016**); 16.5–25.4 nM in (**Zhao et al., 2013d**); 8.3–12 nM in (**Zhao et al., 2012**). [#] Value from (**Zhao et al., 2012**).

DOI: https://doi.org/10.7554/eLife.32056.007

The homo-dimerization $K_D$ of GluA1 and GluA4 is in a range that provides correspondingly high fluorescence signals at concentrations that produce mixed populations of monomers and dimers. This exposes more details in terms of the kinetic stability of homo-dimers in the sedimentation patterns compared to data from titration series experiments which use low nM concentrations of a fluorescent label. The source of this information can be visually discerned in the $c(s)$ overlays of *Figure 4A/B* when comparing the concentration dependence of the dimer peak position: For GluA1 (*Panel A*) the position of the peak shows little change over the range 0.1–1200 nM, and only the relative amplitude changes with concentration. This is characteristic for a monomer-dimer equilibrium with slow dissociation on the time-scale of sedimentation (*Dam et al., 2005*; *Schuck and Zhao, 2017*), implying off-rate constants $k_{off}$ in the range of $10^{-4}\text{sec}^{-1}$ or below. By contrast, for GluA4, the peak position shifts to intermediate values at concentrations of 50 and 300 nM — similar to the behavior observed previously for GluA2 (*Zhao et al., 2013d*) — which is indicative of fast interconversion of species on the time scale of sedimentation (*Gilbert and Jenkins, 1956*; *Dam et al., 2005*; *Schuck and Zhao, 2017*). Shifts in $c(s)$ peak positions are seen also for KaiRs (*Figures 1* and *3*), but the interpretation in this case is obscured by the lack of resolution at the lower signal/noise ratio at fluorophore concentrations of only 0.2–1 nM, where the most parsimonious distribution calculated by maximum entropy regularization in $c(s)$ commonly results in broad peaks for both slow and fast kinetics (*Schuck, 2016*).

To obtain more quantitative estimates of the dissociation rate constants, direct fits of the FDS-SV data with the partial differential equation for sedimentation/diffusion/chemical reaction processes (the Lamm equation) were carried out (*Dam et al., 2005*; *Schuck and Zhao, 2017*). In addition to a high signal/noise ratio for concentrations in the range of the $K_D$, this approach requires complex lifetimes on the timescale of the sedimentation experiment. For GluA4, as well as previously reported data for GluA2 (*Zhao et al., 2013d*) and GluA3 (*Zhao et al., 2012*), only a lower limit of $k_{off} > 3 \times 10^{-3}\text{sec}^{-1}$ can be obtained, which agrees well with the experimentally determined values of $5.1 \times 10^{-3}\text{sec}^{-1}$ and $6.1 \times 10^{-3}\text{sec}^{-1}$ measured by FRET for EGFP and DyLight 488 GluA2, respectively (*Zhao et al., 2013d*). For GluA1, an estimate of $k_{off} = 3 \times 10^{-4}\text{sec}^{-1}$ (95% CI, $1–5 \times 10^{-4}\text{sec}^{-1}$) could be determined. Together with the measured affinities, these values imply lower limits for the on-rate constants $k_{on} > 1.4 \times 10^5 \text{M}^{-1}\text{sec}^{-1}$ for GluA2, contrasting with a value of only $\sim 10^4 \text{M}^{-1}\text{sec}^{-1}$ for GluA1.

## Hetero-association of AMPA receptor ATDs

To examine the assembly of AMPA receptor heterodimers we exploited a titration series strategy to monitor the oligomeric state of the fluorophore carrier as a function of concentration of its unlabeled binding partner. To facilitate accurate analysis of what we expected to be high affinity interactions, with dissociation constants in the nM range, for all AMPA receptor ATD pairs examined, a 0.5 or 1 nM concentration of the fluorescently labeled construct was mixed with the unlabeled binding partner over a wide concentration range. As shown in *Figure 5* the sedimentation coefficient distributions display predominantly monomeric species at low protein concentrations, shifting to predominantly dimeric species at high protein concentrations, indicating nearly full binding saturation. It should be noted that the $s$-values for the monomer, homo-dimer, as well as hetero-dimer species varies for individual pairs, depending on whether the fluorescent label was DyLight 488 (GluA2) or an EGFP fusion (GluA1 and GluA3). Different patterns in the sedimentation profiles observed in these experiments suggests differences in the dissociation kinetics for individual dimer species, but at the low fluorophore concentrations and concomitantly low signal/noise ratios these cannot be unequivocally assigned.

The quantitative analysis of the $s_w$ isotherms obtained after integration of the $c(s)$ distributions is shown in *Figure 6*. Similar to kainate receptors, the hetero-dimerization affinity was determined by fitting a binding model describing the three coupled equilibria for which the $K_D$ and $s$-values for the two homo-dimerization reactions were fixed at pre-determined values described above, while refining the $K_D$ and $s$-value for the heterodimer. The results in *Figure 6A* highlight differences in affinity for heterodimers *vs.* homodimers of GluA2. Favored are complexes with GluA1 and GluA3, with the best-fit $K_D$ of 2.9 nM (95% CI, 1.6–5.6 nM) for GluA1/GluA2, and 1.3 nM (95% CI, 1–1.7 nM) for GluA2/GluA3, whereas the GluA2/GluA4 heterodimer is slightly less stable (33 nM; 95% CI, 19–59 nM) than the homodimer. Independent confirmatory data for the high GluA2/GluA3 affinity was obtained recently using a novel multi-component fluorescence detection technique for SV that could

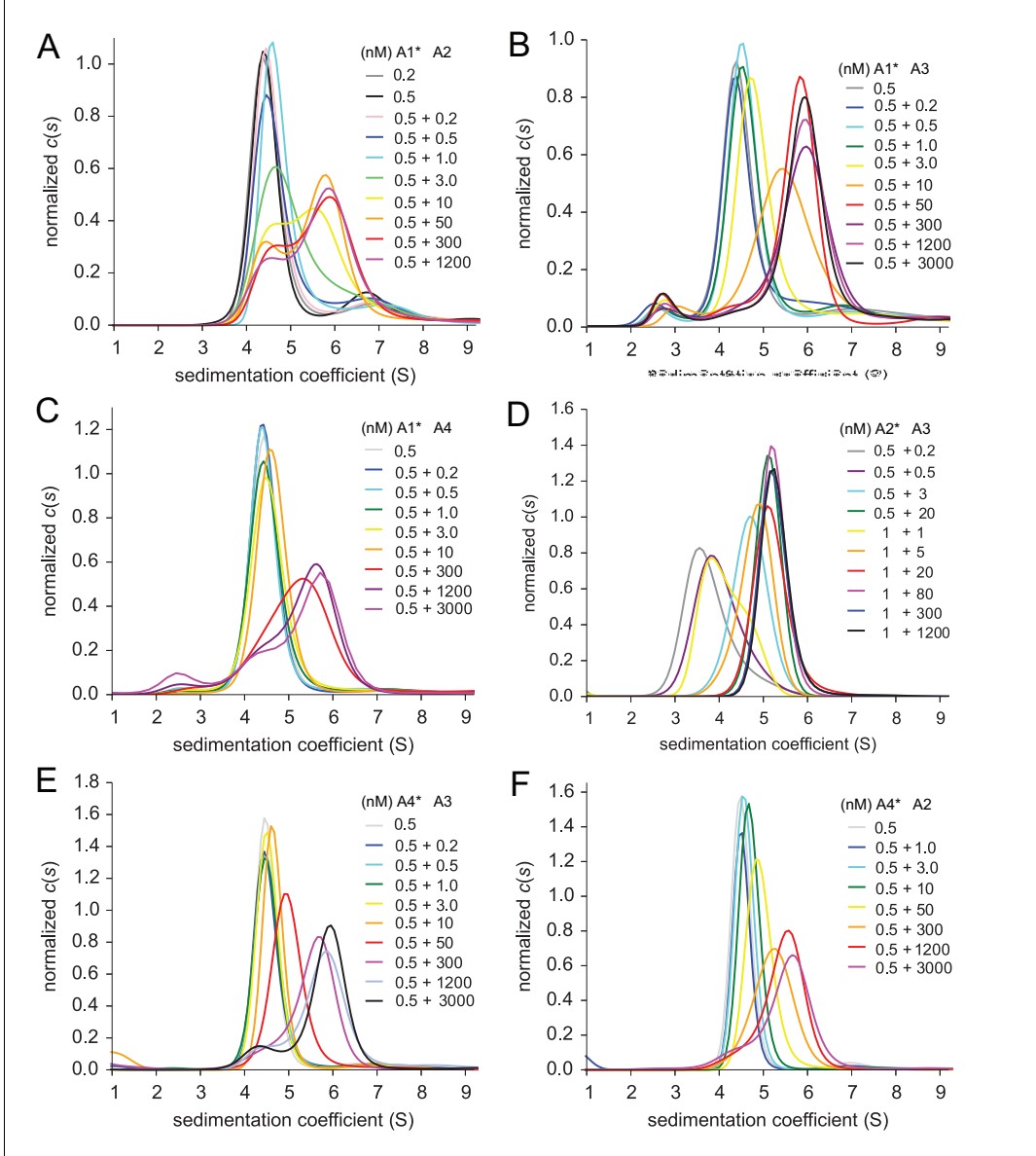

**Figure 5.** Heteromeric assembly of AMPA receptor ATDs. Shown are sedimentation coefficient distributions $c(s)$ for the titrations of 0.2 or 0.5 nM EGFP-GluA1 with 0.2 nM - 1.2 μM unlabeled GluA2 (Panel A), 0.5 nM EGFP-GluA1 1 with 0.2 nM - 3 μM unlabeled GluA3 (Panel B), 0.5 nM EGFP-GluA1 with 0.2 nM - 3 μM unlabeled GluA4 (Panel C), 0.5 or 1 nM DyLight 488-labeled GluA2 with 0.2 nM - 1.2 μM unlabeled GluA3 (Panel D), 0.5 nM EGFP-GluA4 with 0.2 nM - 3 μM unlabeled GluA3 (Panel E), and 0.5 nM EGFP-GluA4 with 1 nM - 3 μM unlabeled GluA2 (Panel F).
DOI: https://doi.org/10.7554/eLife.32056.008

track both GluA2 and GluA3 complexes separately (*Zhao et al., 2016*). As shown in *Figure 6B*, energetically favored assembly of heteromeric species occurs also for AMPA receptor combinations that lack GluA2: For the pairs GluA1/GluA3, the best-fit $K_D$ was 8.3 nM (95% CI, 5.8–12 nM); for GluA1/GluA4 it was 62 nM (95% CI, 16–218 nM); and for GluA3/GluA4 we determined a $K_D$ of 91 nM (95% CI, 60–137 nM).

## Discussion

In the present work we have undertaken a systematic survey of homomeric and heteromeric kainate and AMPA receptor ATD dimer stabilities. On one hand, this can aid the interpretation of an increasing body of iGluR structural data and their dimerization interfaces (*Rossmann et al., 2011*;

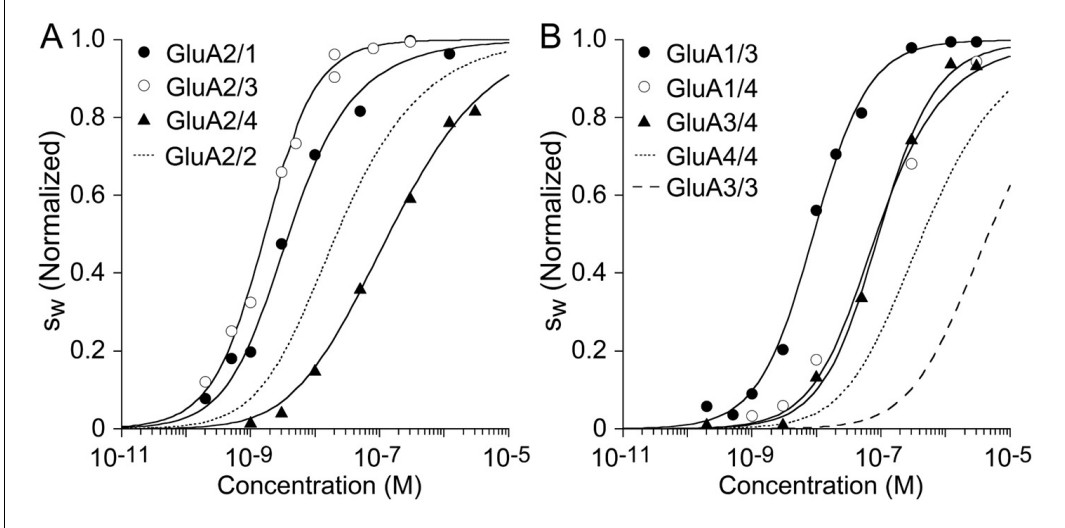

**Figure 6.** Isotherms for heteromeric assembly of AMPA receptor ATDs. Weighted-average $s_w$ as a function of concentration were extracted from the sedimentation coefficient distributions shown in **Figure 5**, and fit with a competitive homo- and hetero-dimerization model using previously determined homo-dimerization parameters of both components as fixed constraints (solid line). Panel A: The analysis of complexes of GluA2 yielded best-fit heterodimer $K_D$-values of 2.9 nM (95% CI: 1.6–5.6 nM) for GluA2/GluA1; 1.3 nM (95% CI: 1.0–1.7 nM) for GluA2/GluA3; and 32.6 nM (95% CI: 19–59 nM) for GluA2/GluA4. For comparison, the dotted line shows the isotherm for homomeric assembly of the GluA2 ATD. (Panel B:) The analysis of heterodimer assemblies not containing GluA2 yields best-fit heterodimer $K_D$-values of 8.3 nM (95% CI: 6–12 nM) for GluA1/GluA3; 61.5 nM (95% CI: 15–218 nM) for GluA1/GluA4; and 91 nM (95% CI: 60–137 nM) for GluA3/GluA4. For comparison, the dashed line shows the isotherm for self-association of GluA3 ATD and the dotted line that of GluA4 ATD. $s_w$ isotherms are normalized to assign 1.0 for the range from monomer to dimer $s$-value.

DOI: https://doi.org/10.7554/eLife.32056.009

*Mayer, 2016*); on the other hand, knowledge of the different energetic stabilities is a reference point for the interpretation of the physiologically observed repertoire and local abundances of various homo- and heterodimers expressed under different patterns of activity in different neurons, dictating synaptic gating kinetics and ion conductivity. The relative free energy of binding of purified receptor ATDs was hypothesized to reconstruct an assembly pathway and reflect the diversity in dimeric populations found in vivo (*Rossmann et al., 2011*), although precise mechanisms are still unclear, including impact of local copy numbers, and the physico-chemical microenvironment. In addition, interactions mediated by other receptor domains, especially the transmembrane domain (*Gan et al., 2016*; *Greger et al., 2003*), and complex formation with auxiliary proteins (*Jackson and Nicoll, 2011*; *Sheng et al., 2017*) are likely to be important determinants of receptor assembly. However, knowledge of the relative thermodynamic stability of assembly of ATD dimers for different species forms a useful guide for the analysis of additional regulatory contributions. Further, at least at some stage of receptor assembly, without invoking as of yet unknown mechanisms, the formation of heterodimers would be directly competitive to the homodimers of their components, and therefore the comparison of their binding energetics and kinetics is of special interest.

The measured binding energies of AMPA receptors, summarized in **Figure 7**, mirror in many ways physiologically observed preferences, with species of greatest stability being the heterodimers GluA2/GluA3, GluA1/GluA2, and GluA1/GluA3. Of note, the GluA1 and GluA3 heterodimer assemblies with GluA2 exhibit $K_D$-values that are 7-fold and 16-fold lower than that for self-assembly of GluA2, indicating that heterodimer assembly is strongly preferred. Formation of GluA2 heterodimers would also be facilitated considering the relatively short life-time of GluA2 homodimers observed in the present work, lowering the energy barrier between different states, although more complete information on other pairs — requiring FDS-SV strategies with higher signal/noise ratio or future experiments with other biophysical techniques — would be necessary to fully understand this aspect. Interestingly, the pairs GluA1/GluA3 and GluA3/GluA4 form heterodimer assemblies of comparable strength to those containing GluA2, indicating that high affinity heterodimer assembly is not limited to AMPA receptor assemblies containing GluA2. For the pairs GluA1/GluA3, the best-fit $K_D$ was 8.3

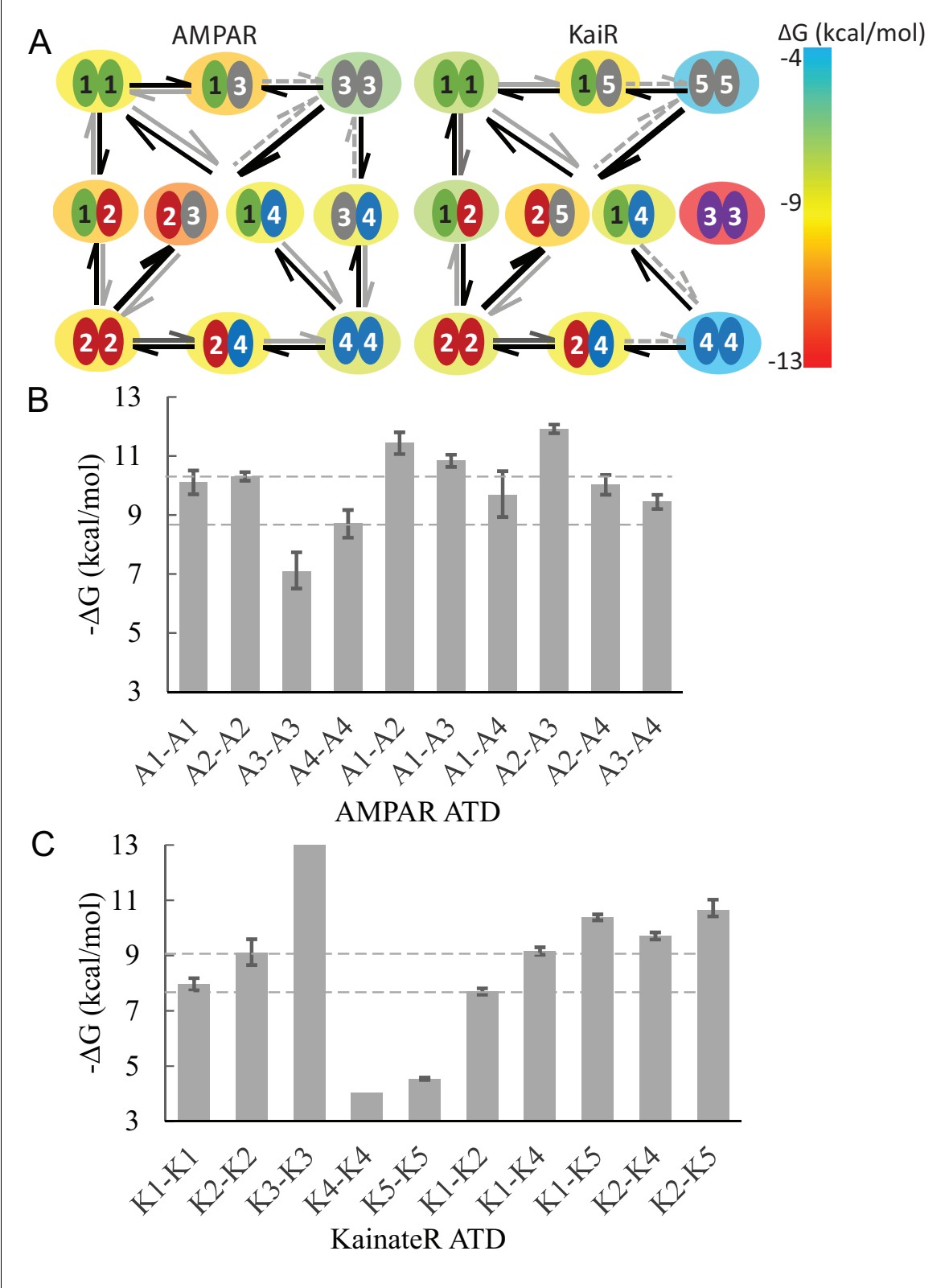

**Figure 7.** Diagram of specific preferential binding of AMPAR and KaiR ATDs. The Gibbs free energy for various dimers at 20°C under the current experimental conditions was calculated from the $K_D$-values in *Table 1* and *Table 2*. Panel A: Graphic presentation of the relationship between the ATDs for AMPAR and KaiR. The color of the big circle (energy cloud) surrounding each pair indicates the free energy level of the dimerization, using the color temperature scale on the right. The width, line type, and color of the arrows emphasizes the stability difference between pairs with exchanged

*Figure 7 continued on next page*

*Figure 7 continued*

subunits. Panel B and C: Bar graphs of the Gibbs free energy for AMPA (**A**) and kainate (**B**) receptor ATD pairs. The dashed lines are visual guides for comparison, at the level of different homodimer free energies.

DOI: https://doi.org/10.7554/eLife.32056.010

nM; for GluA1/GluA4 it was 62 nM; and for GluA3/GluA4 we determined a $K_D$ of 91 nM. These heterodimer assembly $K_D$ values are between 5-fold to 1000-fold lower than for the GluA3 and GluA4 homodimers. In particular, the comparatively low stability of GluA3 homodimer leads to the strongest preference of GluA3 to hetero-dimerize.

Several of the results are in conflict with those obtained in pioneering FDS-SV studies using FAM-labeled preparations (*Rossmann et al., 2011*) which we have shown exhibit a label-induced increase in stability for GluA2 homodimers (*Zhao et al., 2013d*). We have hypothesized this largely to originate from artificial hydrophobic interfaces, which are eliminated in the present work by using DyLight labels carrying charged sulfonate groups, or EGFP fusion constructs (*Zhao et al., 2013d*). Another source of discrepancy arises from data processing not consistently based on rigorous mass transport theory to account for seemingly binding incompetent fractions (*Rossmann et al., 2011*); for example, $c(s)$ distributions exhibiting significant apparent monomer populations are inconsistent with mixtures of molecules at concentrations far above calculated $K_D$ with excess of high-affinity binding partners (*Figure 1* and S2 in *Rossmann et al., 2011*). Similarly to GluA2, in the present work we find ~30-fold lower affinity for GluA4 homodimers and a ~4-fold difference for GluA3 homodimers as compared to results from the FAM-labeled preparations (*Rossmann et al., 2011*). Also, we found similar affinity for GluA1 and GluA2 homodimers, in contrast to 50-fold difference observed from FAM-labeled preparations (*Rossmann et al., 2011*). These differences are qualitatively important, for example, as they resolve the previously reported unexpectedly similar low nM $K_D$-values for the GluA2 homodimer and the hetero-dimer assemblies with GluA1, GluA3 and GluA4 (*Rossmann et al., 2011*), which would impede the formation of GluA2 heterodimers. In this context we note the new kinetic results which for GluA2, GluA3, and GluA4 show at least an order of magnitude shorter homodimer lifetime as compared to GluA1, seemingly facilitating the pathway to form high-affinity heterodimers. Despite the similar homodimer affinities of GluA1 and GluA2, the on-rate constant is at least 15-fold lower for GluA1, which may arise from differences in flexibility and subtle structural differences of the largely conserved homodimer interface comparing the two (*Yao et al., 2011*), and thus may play role in the abundance of GluA1/GluA2 heterodimers.

It is interesting to compare AMPA and kainate receptor ATD dimerization. In general, AMPA receptors bind more tightly for both homo- and hetero-dimerization reactions. Whereas for AMPA receptors heterodimer ΔG estimates range from −9.4 to −11.4 kcal/mol, kainate heterodimers exhibit lower free energies of binding between −7.7 to −10.6 kcal/mol. The strongest KaiR heterodimer assembly is formed by GluK2/GluK5 (best-fit $K_D$ = 9.4 nM), which corresponds well with its physiological identification as a major species in the brain (*Petralia et al., 1994*; *Mulle et al., 1998*; *Contractor et al., 2003*). With the exception of GluK3, all homodimers were found to be relatively weak, which provides larger driving force to form heterodimers among kainate as compared to AMPA receptor ATDs. Especially, the heterodimer $K_D$-values for both GluK4 and GluK5 are more than three orders of magnitude smaller than their homodimer $K_D$-values, consistent with functional experiments which revealed that both GluK4 and GluK5 only form functional receptors when coexpressed with GluK1-3 (*Werner et al., 1991*; *Herb et al., 1992*; *Fisher and Fisher, 2014*). A similar phenomenon was observed for GluN1B/GluN2B NMDA receptor ATD assembly, in which both components alone did not form dimers even at a concentration of ~20 µM and whereas the heterodimer showed a $K_D$ of 0.7 µM, further enhanced by ifenprodil (*Karakas et al., 2011*). This suggests that similar guiding principles of subtype combination may exist for other iGluR families (*Rossmann et al., 2011*). The GluK3 ATD shows an extremely stable homo-dimer with no dissociation observed even in pM regime. Interestingly, the gating properties of GluK3 are also unique with a low affinity for glutamate and Zn modulation that is absent in other KaiRs (*Swanson et al., 1997*; *Veran et al., 2012*). However, co-assembly of GluK3 with GluK2 has been demonstrated (*Cui and Mayer, 1999*), indicating that in vivo hetero-dimer formation is possible in spite of the highly thermodynamically favorable homo-dimer.

Finally, from a methodological point of view, the present work highlights the unique potential of FDS-SV to study complex equilibria of interacting molecules, here including competitive homo- and hetero-associations, over many orders of magnitude of concentrations for systems with pM to µM $K_D$. A powerful feature of FDS-SV is the presentation of the size-distribution of the observed components, and their changes in population or migration patterns in the course of titration series. This provides for independent internal controls for the quality and assembly state of the samples under study, and allows for highly reproducible results (*Table 2*). At the same time, the present work demonstrates the requirement for rigorous analysis and thorough control experiments to eliminate potential pitfalls from attaching fluorescent labels (*Zhao et al., 2013a*). Though not universally applicable, EGFP fusions proteins have the advantage of a well-controlled 1:1 labeling ratio, and do not require laborious and expensive optimization of chemical labeling and associated extra purification steps to remove free dye. The additional mass of EGFP fusion constructs also leads to greater hydrodynamic distinction between homo- and heterodimer species in titration experiments with unlabeled proteins. Furthermore, with minor modifications to introduce photoswitchable variants, it offers opportunities for monochromatic multi-component detection in FDS-SV (*Zhao et al., 2016*). We also found that, if data with mixed populations of species can be obtained with sufficient signal/noise ratio, complex life-times can be measured, which may describe an avenue for more detailed future studies of iGluR assembly pathways. However, for most of the molecules studied here, this will require improved ultracentrifuge fluorescence detectors.

In summary, the present work provides a free energy map of AMPA and kainate receptor ATD dimerization, which describe driving forces for homo- and hetero-dimerization of different receptor subtype combinations. We believe this will contribute to the understanding of the mechanism of assembly of functional receptors, and provides rationales for their observed composition and distribution in vivo on the basis of their molecular properties.

## Materials and methods

### Protein expression and purification

Plasmids for expression of secreted proteins using HEK cell suspension cultures were designed as follows. For unlabeled proteins, and for preparations that were later labeled with DyLight 488, the following cDNAs for iGluR ATDs with native signal peptides were cloned into the pRK5-IRES-EGFP mammalian expression vector: GluK1 G213N/M215T, last residue Arg402; GluK2 G215N/M217T, last residue Lys389; GluK3 T221N/Y223S, last residue Arg392; GluK4, last residue Val399; GluK5, last residue Ile387; GluA2, last residue Ser383; GluA3, last residue Ser386; followed in all constructs with a C-terminal linker, thrombin cleavage site and affinity tag, sequence: LELVPRGS-His8. Where indicated, mutations in the lateral face of ATD dimer assemblies were introduced to insert N-linked glycosylation sites that prevent protein aggregation, as reported previously for GluK2 (*Kumar et al., 2011*). For GluA1 and GluA4 we created N-terminal EGFP A206K mutant fusion proteins as follows. The cDNA for the GluA1 signal peptide and ATD, residues Met1-Phe21, was coupled in frame to an SGSG tetrapeptide followed by EGFP residues Val2-Lys239, followed by a second SGSG tetrapeptide and then the GluA1 ATD cDNA, residues Val17- Gln405, followed by a linker peptide, thrombin cleavage site and affinity tag peptide, sequence: SGLRSGLVPRGS-His8, and cloned into pFastBac1 baculovirus expression vectors for protein expression in Sf9 insect cells. For GluA4, the cDNA for the signal peptide and ATD, residues Met1-Gly21, was coupled in frame to an SGSG tetrapeptide followed by EGFP residues Val2-Lys239, followed by a second SGSG tetrapeptide and then the GluA4 ATD cDNA, residues Ala22- Glu413, followed by a linker peptide, thrombin cleavage site and affinity tag peptide, sequence LVPRGS-His8, and cloned into pRK5 for protein expression in HEK cells. Following expression induced by transient transfection of HEK cell suspension cultures using PEI (*Aricescu et al., 2006*), or for the GluA1-EGFP fusion protein baculovirus infection of insect cell cultures, media were clarified by centrifugation, concentrated by ultrafiltration (Millipore Labscale TFF system, Pellicon Ultracel 10 kDa), loaded onto $Ni^{2+}$ charged HiTrap chelating HP columns (GE Health Care) and eluted with linear imidazole gradients. Pooled fractions were digested with thrombin to remove affinity tags followed by further purification using ion exchange chromatography.

## Protein labeling

Purified concentrated iGluR ATD samples were mixed with N-hydroxysuccinimide (NHS) ester-activated DyLight 488 (Thermo Fisher Scientific) dissolved in DMSO and then resuspended in labeling buffer (150 mM NaCl, 1 mM EDTA, 20 mM $Na_2HPO_4/NaH_2PO_4$ pH 7.0), as reported previously (*Zhao et al., 2013d*). The reactions were incubated in the dark at room temperature; the incubation time was typically 1 hr. The reaction solution was then loaded onto a high resolution size exclusion chromatography column at 4°C (Superdex 75 10/300 GL) equilibrated with labeling buffer at pH 7.5 to separate free dye from labeled protein. The protein concentration and labeling ratio were then determined by UV-Vis spectrophotometry using extinction coefficient values of the unmodified protein calculated from the amino acid sequence, and $\epsilon_{280}$ of 10,290 $M^{-1}cm^{-1}$ and $\epsilon_{493}$ of 70,000 $M^{-1}cm^{-1}$ for DyLight 488. Labeling ratios were typically 0.9–1.3 and final protein concentrations generally 1–5 μM.

## Sedimentation velocity analytical ultracentrifugation (SV-AUC)

Our analysis of ATD dimerization was performed using sedimentation velocity (SV) rather than sedimentation equilibrium (SE) for three reasons. First, SV offers better size-resolution than SE; second, the shorter run time for SV lowers the susceptibility to degradation artifacts which we previously established can impact the study of ATD assembly by SE (*Zhao et al., 2012*); third, SV is better suited for the data structure of fluorescence optical detection (*Zhao et al., 2013c*). Sedimentation velocity experiments were performed in an Optima XL-A analytical ultracentrifuge (Beckman Coulter, Indianapolis, IN) equipped with a fluorescence detection system (FDS) using a 10 mW solid-state laser at 488 nm as the light source (AVIV Biomedical, Lakewood, NJ) or an Optima XL-I analytical ultracentrifuge with an absorbance optical detection system at 280 nm. Concentration series for the homo- and hetero-interactions were prepared with either titration or dilution strategies. In titration series, the protein carrying the fluorophore was kept at a constant low nM concentration in each sample, while the unlabeled binding partner concentration was varying in a wide range up to μM. For some homo-dimerization analyses, a dilution series was employed by diluting the stock solution to the targeted concentrations. In all FDS-SV experiments reported in this study, we included 0.1 mg/mL of bovine serum albumin (BSA) in the samples in order to minimize surface adsorption of the protein of interest in the AUC cell.

Prior to the SV experiments, standard protocols were followed as described in (*Zhao et al., 2013b*; *Chaturvedi et al., 2017a*). The protein samples were loaded into AUC cell assemblies with standard charcoal-filled Epon double-sector centerpieces with 12 mm pathlength and quartz or sapphire windows. The AUC cell assemblies filled with samples were subject to at least 3 hr temperature equilibration at 20°C before initiation of the SV experiments. No concentration-dependent change in quantum yield was observed (*Chaturvedi et al., 2017b*). In general, a focusing depth at 3955 μm for the laser beam was used. Depending on the brightness and the concentrations of the fluorescent molecule, the PMT setting varied to ensure an optimal fluorescent intensity below 3000 counts per sec. Generally PMT voltage was set at 30–50%. For the samples with pM concentration fluorescent molecules, usually 80% PMT voltage was used to maximize the sensitivity of the detection. After temperature equilibration, the rotor loaded with the samples was accelerated to 50,000 rpm and continuous data acquisition was started immediately. In order to account for the signal contribution of the carrier protein and buffer, a cell loaded with working buffer and 0.1 mg/mL of BSA in the two sectors respectively was included in the experiments. The signal from these two control samples was examined. In experiments using a high photomultiplier voltage necessary for detecting very low fluorophore concentrations, the signal from BSA was significant and was subtracted from the data (*Schuck et al., 2015*; *Chaturvedi et al., 2017a*). When a lower photomultiplier voltage was used, this signal was usually lower than the noise of the data, and thus could be ignored.

## Data analysis

### Sedimentation coefficient distribution and binding isotherm analysis

The SV scans of each sample were loaded into SEDFIT (sedfitsedphat.nibib.nih.gov) and $c(s)$ analysis was applied to resolve the diffusion-deconvoluted sedimentation coefficient distribution ($c(s)$ distribution) of non-interacting species adhering to the hydrodynamic scaling law of compact particles, using maximum entropy regularization, as described (*Schuck, 2000*). For each concentration series,

the resulting $c(s)$ distributions were loaded into the software GUSSI (**Brautigam, 2015**) for superimposed plotting and integration across the $s$-value range of monomeric and dimeric species to determine the signal-weighted average sedimentation coefficient $s_w$ of the corresponding samples, corrected for BSA contributions where necessary (see above). In the present study, no corrections to standard conditions were carried out, and the $s$-values shown in all $c(s)$ and isotherm plots are for experimental conditions. The corrected $s_w$ values were assembled into an isotherm as a function of the protein concentrations, and loaded in SEDPHAT (sedfitsedphat.nibib.nih.gov) for modeling in accordance with the transport method (**Schuck and Zhao, 2017**) with homo- and/or hetero-dimerization models. For the application of the transport method, which is a consideration of mass balance of sedimentation, the only requirement is a good fit of the $c(s)$ sedimentation model to the experimental data, which was achieved without exception.

For homodimerization, mass action law between monomeric species at molar concentration $c_{X1}$ and dimeric species at molar concentration $c_{X2}$ of component X was imposed

$$c_{X2} = K_{X,12}c_{X,1}^2 \ ,$$

(1)

with equilibrium association constant $K_{X,12} = 1/K_{D,X}$, leading to contributions to the observed $s_w$ of

$$s_{w,X} = \frac{s_{X,1}c_{X,1} + 2s_{X,2}c_{X,2}}{c_{X,1} + 2c_{X,2}} \ ,$$

(2)

where $s_{X,1}$ and $s_{X,2}$ denote monomeric and dimeric species' $s$-values, respectively. These were treated as parameters to be refined in the isotherm of component X, alongside $K_{X,12}$. If the $s$-values of monomer and dimer are well-determined by $c(s)$ distributions of slowly interacting systems, they can be fixed to the concentration-independent peak $c(s)$ values (**Schuck and Zhao, 2017**). This allows a dimerization-incompetent fraction to be identified from the $s_w$ isotherm in addition to $K_{X,12}$ with little parameter correlation (**Schuck and Zhao, 2017**). The need for this additional consideration arose for GluA1 and GluA4, recognized from a concentration-independent monomer fraction even at high concentrations saturating binding (**Schuck and Zhao, 2017**); incompetent fractions varied for individual preparations from 0–23% and 0–33%, respectively. For all other molecules, the monomer and dimer $s$-values extrapolated as the asymptotic endpoints of the $s_w$ isotherm were consistent with peak $c(s)$-values for monomeric and dimeric populations, respectively, thus not requiring consideration of binding-incompetent fractions.

For mixtures of components X and Y, hetero-dimerization was described with the equilibrium relationship

$$c_{XY} = K_{XY}c_{X,1}c_{Y,1} \ ,$$

(3)

with equilibrium association constant $K_{XY} = 1/K_{D,XY}$, to be fulfilled in addition to the homo-dimerization of X and Y, each independently following (**Equation 1**) with their respective homo-dimerization constant $K_{X,12}$ and $K_{Y,12}$, and satisfying mass conservation $c_{Y,tot} = c_{Y,1} + 2c_{Y,2} + c_{XY}$, rendering the homo- and hetero-dimerization processes competitive. While the total protomer concentration of the labeled component $c_{X,tot}$ was determined from the observed signal, $c_{Y,tot}$ of the unlabeled component was taken as prior knowledge from the known loading concentration. The observed $s_w$ of the mixture tracing species X was modeled with contributions from the two coupled homo-dimerization and the hetero-dimerization processes

$$s_{w,XY} = \frac{s_{X,1}c_{X,1} + 2s_{X,2}c_{X,2} + s_{XY}c_{XY}}{c_{X,1} + 2c_{X,2} + c_{XY}}$$

(4)

with predetermined parameters from the homo-dimerization and refined parameters in the least-squares fit only $K_{XY}$ and $s_{XY}$. This fit to experimental data was carried out in SEDPHAT as a special case of the general homo/heterodimerization model switching off tetramerization. Confidence intervals were determined using error projection analysis in SEDPHAT to determine 95% confidence level (95% CI) based on F-statistics (**Johnson and Straume, 1994**).

## Determination of the complex lifetime with Lamm equation modeling

Alternative to examining the effect of homo- and hetero-dimerization through its effect on the overall mass transport, which is achieved in the modeling of the $s_w$ isotherm as a function of loading concentrations, it is possible to directly fit the evolution of experimental boundary profiles with the explicit partial differential equation for the sedimentation/diffusion/reaction process (*Stafford and Sherwood, 2004*; *Dam et al., 2005*; *Schuck and Zhao, 2017*), replacing the equilibrium relationships of the coupled homo- and heterodimer formation *Equations 1 and 3* with their corresponding chemical rate equations, in combination with the description of concentration gradients from sedimentation and diffusion by the Lamm equation (*Lamm, 1929*; *Fujita, 1975*). The advantage of this approach is that it can extract information on the kinetics of the reaction from the detailed boundary shapes, but only under the premise of highly pure components and for data of sufficiently high signal/noise ratio, due to the exquisite sensitivity of the boundary shapes for micro-heterogeneity and the limited kinetic range for which sedimentation profiles are sensitive to chemical kinetics (*Dam et al., 2005*; *Brautigam, 2011*; *Schuck and Zhao, 2017*). These conditions were met for some of the samples studied, and for these data, the SV profiles were directly modeled with numerical solutions to the coupled Lamm equation embedding the dimerization binding model with $s$-values for monomer and dimer fixed at the values determined from $s_w$ isotherm analysis, refining the dissociation kinetic rate constant, $k_{off}$ along with $K_D$, by direct fitting of sedimentation profiles in SEDPHAT.

## Acknowledgements

This work was supported by the intramural research programs of NIBIB and NICHD, National Institutes of Health, Bethesda, USA.

## Additional information

### Funding

| Funder | Grant reference number | Author |
| --- | --- | --- |
| National Institutes of Health | Z01-HD000707-31 | Mark L Mayer |
| National Institutes of Health | ZIA EB000008-11 | Peter Schuck |

The funders had no role in study design, data collection and interpretation, or the decision to submit the work for publication.

### Author contributions

Huaying Zhao, Conceptualization, Data curation, Formal analysis, Validation, Investigation, Visualization, Methodology, Writing—original draft, Writing—review and editing; Suvendu Lomash, Sagar Chittori, Resources, Investigation, Writing—review and editing; Carla Glasser, Resources, Investigation; Mark L Mayer, Conceptualization, Supervision, Funding acquisition, Validation, Investigation, Visualization, Writing—original draft, Writing—review and editing; Peter Schuck, Resources, Software, Funding acquisition, Validation, Investigation, Methodology, Writing—original draft, Writing—review and editing

### Author ORCIDs

Sagar Chittori  https://orcid.org/0000-0003-1417-6552
Mark L Mayer  http://orcid.org/0000-0003-4378-8451
Peter Schuck  http://orcid.org/0000-0002-8859-6966

### Decision letter and Author response

Decision letter https://doi.org/10.7554/eLife.32056.012
Author response https://doi.org/10.7554/eLife.32056.013

## Additional files

**Supplementary files**

• Transparent reporting form
DOI: https://doi.org/10.7554/eLife.32056.011

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
