## [Decision Letter]

Thank you for submitting your article "Preferential Assembly of Kainate and AMPA Receptor Amino Terminal Domains" for consideration by *eLife*. Your article has been reviewed by three peer reviewers, and the evaluation has been overseen by Gary Westbrook as the Senior Editor. The reviewers have opted to remain anonymous. The reviewers have discussed the reviews with one another and the Reviewing Editor has drafted this decision to help you prepare a revised submission.

Summary:

The reviewers all recognized the high technical quality and comprehensive nature of the experiments that can serve as a template or building block for future studies of ionotropic glutamate receptor structure and assembly. Of course, isolated ATDs cannot tell us everything, but neither can structures of membrane receptors with truncated C-termini, various mutations to stabilize domains etc. Thus, the authors should be explicit in the revised manuscript about the limitation of interpretations from their results, particularly with respect to additional factors that may play a role in receptor assembly. You will see this issue reflected in the comments particular of reviewers 1 and 3. Please also address the specific concerns noted in the reviews below.

Reviewer #1:

Glutamate receptors mediate the majority of excitatory neurotransmission in the central nervous system (CNS), are important for high cognitive brain functions and represent targets for treatment of numerous neurological diseases. The Zhao et al. manuscript presents a high-quality study of AMPA and kainate subtype glutamate receptor ATD dimerization properties using an advanced AUC technique. The results are solid and to a great extent supplement and in some instances correct previous measurements. No doubt, this study will make a great reference for future work on assembly of ionotropic glutamate receptors.

The major caveat of this work is a possibly wrong assumption about importance of ATD dimer interface stability for glutamate receptor assembly. While this stability might indeed serve as a driving force for the initial steps of assembly, the entire process ends up producing a structure with many more interfaces, including the larger size interfaces in the transmembrane region. Any of these additional interfaces might be potentially more important for the overall assembly of glutamate receptors. As rightfully mentioned by the authors, the issue is even more complicated because of the presence of glutamate receptor auxiliary subunits that seem to accompany the majority of receptors in the CNS synapses. To properly address glutamate receptor assembly, the approach used in the present study would need to be applied to full length glutamate receptors or their complexes with auxiliary subunits.

Additionally, while this work is important for assessing interaction of ATDs as possible important factors at the initial steps of glutamate receptor assembly, similar measurements have been previously conducted by this same group as well as others. Conceptually, the binding diagram in Figure 7 of this manuscript represents similar results and ideas as Figure 7 in 2011 paper by Rossmann et al.

Reviewer #2:

The manuscript by Zhao et al. reports the work related to the biochemical/molecular mechanism underlying the preferential assembly of heteromeric AMPA and kainate receptors. Heteromeric oligomerization of non-NMDA receptors (AMPA and kainate receptors) is a fundamental biological question since native non-NMDA receptors are known to exist as heterotetramers. However, patterns of heteromeric subunit association remains obscure. It's been known that an amino terminal domain (NTD) of ionotropic glutamate receptors is critical in subunit assembly. The current study addresses the question regarding the pattern of subunit assembly by isolating the purified NTD proteins from various subunits from both kainate and AMPA receptor families and measure their association constants by a series of analytical ultracentrifugation (AUC) experiments. All of the AUC experiments are conducted at the highest level with the most updated technology involving fluorescence labeling and the fluorescence detection to allow detection of the NTD proteins at low concentrations and precise measurement of high-affinity subunit association, which would not be possible if conventional detection methods such as absorbance and interference are used. The conclusion in Figure 7 is complete and translates well to the biology of ionotropic glutamate receptors. The methodological details in this work is also meaningful to the community of AUC. The manuscript, in general, reads very well.

–Why did the authors choose sedimentation velocity at various protein concentrations over sedimentation equilibrium? An explanation for this choice would be meaningful for readers.

–Abstract – The sentence “Kainate receptor ATD dimers are generally lower affinity than [...]” would sound strange to the general reader. The association of ATD dimers are stronger or weaker.

Reviewer #3:

This manuscript describes a technically challenging biochemical analysis of the binding energies for the isolated amino terminal domains of the AMPA and kainate receptors. I have no concerns about the technical execution of the experiments, the analytical approach, or the rigor of the study. The results support some previously published work and extends our understanding of amino terminal domain binding in a comprehensive fashion across the full family of AMPA and kainate receptors. Interestingly, the authors find a range of affinities and thus lifetimes for dimer formation, and suggest, on the assumption that this is the key event driving assembly, that this heterogeneity will influence or perhaps control both receptor distributions in neurons as well as influence ability of neurons to change subunit composition. The information is certainly valuable, particularly as a systematic and comprehensive approach to evaluating all subunits. The authors argue they have circumvented technical artifacts associated with previous studies, and in this sense the comprehensive approach creates a data set that can be uniformly relied on by interested individuals. This is far more valuable than piecing together a story in which some subunit interactions were evaluated by one group, and others by a different group using different approaches. True to the authors' publication history, this is an excellent study of a molecular question that is important for basic understanding of receptor structure and assembly. I have no specific suggestions on ways to improve the study nor do I have any suggested wording changes that might improve clarity.

The key question is whether this uniform dataset is impactful and influential in directing others to new insight, and whether it will stimulate work in the field that will advance our understanding. On one hand, the authors make a compelling case that this information "can aid the interpretation of an increasing body of iGluR structural data […]", and I agree with this statement. They also state that "knowledge of the relative thermodynamic stability of the different species may guide the analysis of regulatory contributions", which I also agree with. I also believe that rigorous data available in a comparative manner can be influential by allowing future dissection of ideas that might rely, for example, on different ATD stability; such analysis would not be possible without the comprehensive approach taken, so in this way the study, because of its rigor, can guide and stimulate future work.

On the other hand, I believe it is perhaps a stretch to assume the affinities describe here will serve as a "reference point for interpretation of physiologically observed repertoire and local abundances of various homo- and heterodimers expressed under different patterns of activity […]". I do not see this work influencing physiologists, nor do I anticipate ATD affinity to be the over-riding driving force dictating which receptors that reach the cell surface. The authors admit as much a few sentences later saying local copy number, the microenvironment, other protein domains, and auxiliary subunits will all influence assembly. On balance, I favor publication because these comprehensive studies can stimulate thinking about new questions in ways partial evaluations do not.

---

## [Author Response]

Reviewer #1:[…] The major caveat of this work is a possibly wrong assumption about importance of ATD dimer interface stability for glutamate receptor assembly. While this stability might indeed serve as a driving force for the initial steps of assembly, the entire process ends up producing a structure with many more interfaces, including the larger size interfaces in the transmembrane region. Any of these additional interfaces might be potentially more important for the overall assembly of glutamate receptors. As rightfully mentioned by the authors, the issue is even more complicated because of the presence of glutamate receptor auxiliary subunits that seem to accompany the majority of receptors in the CNS synapses.

We thank the reviewer for the comment that “No doubt, this study will make a great reference for future work on assembly of ionotropic glutamate receptors”. We agree that the importance of ATD dimerization compared to other receptor domains is not entirely clear, but believe that the data presented in the manuscript will help to evaluate to what extent the driving force of ATD dimerization can account for observed assemblies. It is important to note that in the few cases where mutations in the ATD dimer interface that lower heterodimer affinity have been studied using electrophysiological techniques, the effects observed suggest that the ATD does play a major role in receptor assembly in vivo, as reported in Kumar et al., 2011 and Rossmann et al., 2011 We have added additional text and citations indicating that other regions, most notably the transmembrane domain, play important roles in receptor assembly, extending text in the original submission that already drew attention to this issue.

To properly address glutamate receptor assembly, the approach used in the present study would need to be applied to full length glutamate receptors or their complexes with auxiliary subunits.

From a technical perspective, it is impossible to extend the approach used in the present study to full length glutamate receptors for multiple reasons. Use of AUC to measure the strength of interactions between different subunit combinations would require that detergent solubilized full length iGluRs reversibly dissociate into monomer and dimer assemblies; this has never been demonstrated, and instead tetramer dissociation leads to denaturation and aggregation.

Additionally, while this work is important for assessing interaction of ATDs as possible important factors at the initial steps of glutamate receptor assembly, similar measurements have been previously conducted by this same group as well as others. Conceptually, the binding diagram in Figure 7 of this manuscript represents similar results and ideas as Figure 7 in 2011 paper by Rossmann et al.

Previous studies were either incomplete, and did not examine all subunit combinations, or were technically flawed. The results reported in our *eLife* submission are the first comprehensive survey of kainate receptor ATD assembly; our prior work was limited to GluK1, GluK5 and their heterodimer assembly, studied using conventional absorbance optics. There is no data in the literature for GluK1, GluK3, GluK4 and their interactions with GluK2 and GluK5. For AMPA receptors, we previously used and GluA2 and GluA3 as model systems to develop techniques which overcame limitations in the study by Rossmann et al., 2011 that resulted from use of a fluorescent label that perturbed ATD assembly, and to establish more advanced fluorescence techniques for *eLife*.

Reviewer #2:[…] Why did the authors choose sedimentation velocity at various protein concentrations over sedimentation equilibrium? An explanation for this choice would be meaningful for readers.

We have added text explaining that sedimentation equilibrium studies of systems with high affinity association are exquisitely sensitive to contamination by low MW contaminants and can be impacted by extremely minor proteolytic degradation during the typically days long runs required to reach equilibrium at multiple rotor speeds. As we reported previously, this was found to occur during SE studies of GluA2 in our own lab, and was very likely the cause of low affinity K_D_s reported by two other laboratories for AMPA receptor ATDs also determined by SE. While the susceptibility to trace degradation products could be lessened with the use of lower concentrations with the fluorescence optics, the feasibility of sedimentation equilibrium measurements for dimer assembly with fluorescence optics has yet to be established. By contrast, SV experiments avoid this issue because individual species are resolved during the run, and in addition, the runs typically require only a few hours of data collection.

Abstract – The sentence “Kainate receptor ATD dimers are generally lower affinity than […]” would sound strange to the general reader. The association of ATD dimers are stronger or weaker.

We have changed the text to “The association of kainate receptor dimers is generally weaker than AMPA receptor dimers […]”

Reviewer #3:[…] The key question is whether this uniform dataset is impactful and influential in directing others to new insight, and whether it will stimulate work in the field that will advance our understanding. On one hand, the authors make a compelling case that this information "can aid the interpretation of an increasing body of iGluR structural data…", and I agree with this statement. They also state that "knowledge of the relative thermodynamic stability of the different species may guide the analysis of regulatory contributions", which I also agree with. I also believe that rigorous data available in a comparative manner can be influential by allowing future dissection of ideas that might rely, for example, on different ATD stability; such analysis would not be possible without the comprehensive approach taken, so in this way the study, because of its rigor, can guide and stimulate future work.On the other hand, I believe it is perhaps a stretch to assume the affinities describe here will serve as a "reference point for interpretation of physiologically observed repertoire and local abundances of various homo- and heterodimers expressed under different patterns of activity…" I do not see this work influencing physiologists, nor do I anticipate ATD affinity to be the over-riding driving force dictating which receptors that reach the cell surface. The authors admit as much a few sentences later saying local copy number, the microenvironment, other protein domains, and auxiliary subunits will all influence assembly. On balance, I favor publication because these comprehensive studies can stimulate thinking about new questions in ways partial evaluations do not.

We are in agreement with most of the points raised by the reviewer. Specifically, as noted in the manuscript, other regions, especially the transmembrane domain, must be major determinants of assembly. How this is regulated is currently a complete mystery. Possibly chaperones permit subunit exchange between dimer assemblies of full length iGluRs before the tetramer is assembled, and in this scenario competitive, affinity driven interactions between ATD homo and heterodimers could play an important role. We hope that our study, like that reported by Rossmann et al., 2011, will stimulate physiologists to consider the mechanisms that control glutamate receptor subunit composition in vivo, although the reviewer seems dubious. We agree with the reviewer that the availability of a comprehensive and reliable data basis is most important at the current state of the field, so as to allow critical evaluation of mechanistic theories.